# Views and experiences of young people on using mHealth platforms for sexual and reproductive health services in rural low-and middle-income countries: A qualitative systematic review

**Alexander S. Laar* , Melissa L. Harris, Md N. Khan, Deborah Loxton**

The University of Newcastle, Australia, School of Public Health and Medicine, Centre for Women's Health Research, Faculty of Health and Medicine, Hunter Medical Research Institute, Callaghan, New South Wales 2308, Australia

* Alexander.Laar@uon.edu.au

**Data Availability Statement:** The datasets used and/or analysed during the current study are provided as Supporting Information.

## Abstract

In low- and middle-income countries (LMICs), reproductive health programs use mobile health (mHealth) platforms to deliver a broad range of SRH information and services to young people in rural areas. However, young people's experiences of using mobile phone platforms for SRH services in the rural contexts of LMICs remains unexplored. This review qualitatively explored the experiences and perceptions of young people's use of mobile phone platforms for SRH information and services. This qualitative evidence synthesis was conducted through a systematic search of online databases: Medline, Embase, CINAHL, PsycInfo and Scopus. We included peer reviewed articles that were conducted between 2000 to 2023 and used qualitative methods. The methodological quality of papers was assessed by two authors using Grading of Recommendations, Assessment, Development and Evaluation (GRADE) and Confidence in Evidence from Reviews of Qualitative research (CERQual) approach with the identified papers synthesized using a narrative thematic analysis approach. The 26 studies included in the review were conducted in a wide range of LMIC rural settings. The studies used seven different types of mHealth platforms in providing access to SRH information and services on contraception, family planning, sexually transmitted infections (STIs) and human immunodeficiency virus (HIV) education. Participant preferences for use of SRH service platforms centred on convenience, privacy and confidentiality, as well as ease and affordability. High confidence was found in the studies preferencing text messaging, voice messaging, and interactive voice response services while moderate confidence was found in studies focused on phone calls. The overall constraint for platforms services included poor and limited network and electricity connectivity (high confidence in the study findings), limited access to mobile phones and mobile credit due to cost, influence from socio-cultural norms and beliefs and community members (moderate confidence in the study findings), language and literacy skills constraints (high confidence in the study findings). The findings provide valuable information on the preferences of

**Funding:** The author(s) received no specific funding for this work.

**Competing interests:** The authors have declared that no competing interests exist.

mHealth platforms for accessing SRH services among young people in rural settings in LMICs and the quality of available evidence on the topic. As such, the findings have important implications for health policy makers and implementers and mHealth technology platform developers on improving services for sustainable adoption and integration in LMIC rural health system.

## Author summary

This paper used the GRADE-CERQual (Confidence in Evidence from Reviews of Qualitative research) approach developed by the GRADE (Grading of Recommendations Assessment, Development and Evaluation) Working Group. We adapted GRADE-CERQual approach exploring the views and experiences of young people on using mHealth platforms for sexual and reproductive health services in rural low-and middle-income countries (LMICs).

Our review provides an innovative approach on applying the GRADE-CERQual guideline for the assessment of confidence in a review finding. The study findings reported preferences and barriers for use of mHealth-based platforms for SRH information and services among rural young people from rural LMICs. The findings show that mHealth technology has the potential to address numerous SRH information and service access barriers. We provide relevant recommendations for policy makers and advisors of SRH programs to improve mHealth service delivery in the rural health areas of LMICs.

## Introduction

Sexual and reproductive health (SRH) is essential to young people's overall health and wellbeing globally [1,2]. While the 1994 Conference on Population and Development recognised the rights of young people to access SRH services, including contraception and family planning [2], in low- and middle-income countries (LMICs) most young people aged 10–24 years have very limited or no access to SRH information and services. Further, young people from rural settings in LMICs remain the most underserved population by SRH information and services [3–7].

Studies from rural LMICs have identified multiple challenges which limit young people's access to and use of SRH information and services [5,8–10]. Some of these barriers include socio-cultural norms [11], stigma attached to sex and sexual health by communities [11], and religious organisations [11]. Socio-cultural norms and parental influence play significant role in restricting the use of conventional SRH information services among population in rural LMICs especially young people [12–14]. Also, socio-cultural norms and parental factors are found to have potential impact on the uptake of mobile health (mHealth) SRH services among young people in rural contexts in LMICs due to restriction of mobile phone use among young people by parents and other significant community members [15,16].

In addition, challenges such as limited health resources and infrastructure, negative or unprofessional attitudes of health providers, long distances to health facilities, inability to afford cost of services [11] and long waiting times for services at health facilities deter or prevent young rural people from accessing services [4–6]. Additionally, challenges of conventional SRH intervention services results in limited access to SRH [4,17,18]. These barriers contribute to an unmet need for SRH information among young rural people and result in

non-use or incorrect use of contraception and expose young people to vulnerabilities and risks which may lead to unintended pregnancies and STIs including HIV [2,19,20].

Meeting young people's SRH information and service needs, especially in rural and remote areas, will require innovative modes of delivering youth-friendly services. Digital health programs offer opportunities to improve the provision and access to SRH of young people in LMICs [21]. Using mobile phones for the delivery of SRH information and services in a rural context of LMICs could reduce or avert inequalities in the uptake of SRH information and services among young people in rural settings. To address the specific SRH needs of young people in remote rural settings (i.e. in hard-to-reach areas), novel and innovative mobile health (mHealth) platforms are now being used in LMICs to provide access to youth-friendly and youth-centred SRH information and services [22]. With the rapid evolution in mHealth platforms interventions for SRH services for the past two decades in LMICs is rapidly helping to address conventional access barrier and challenges especially among young people in remote rural settings [6,23,24]. Innovative mHealth initiatives have the potential of providing targeted, tailored, and accessible youth friendly SRH information and services based on young people needs and preferences [6,23,24].

The World Health Organization defines mHealth as 'the use of mobile and wireless technologies to support the achievement of health objectives' [22]. Digital health technologies such as mobile phones can facilitate SRH information exchange and use among young people in rural LMICs to assist with making informed SRH decisions [22]. It is suggested that these technologies are effective and affordable in improving coverage and access to SRH information and services among young rural people and are acceptable to young people [6,22].

However, despite the potential of mHealth platforms to improve provision and access to SRH information and services in rural settings in LMICs [22,25], little evidence and information currently exists. What is available is qualitative in nature, which limits generalisability. A synthesis of this literature, however, can be undertaken to inform policy programs designed to address the SRH needs of young people in rural healthcare systems in LMICs. As such, we synthesised the current evidence to provide a comprehensive understanding of the views and experiences of young people in rural areas using mobile platforms for SRH information and services in rural and remote areas in LMICs. This study qualitatively explored the experiences and perceptions of young people's use of mobile phone platforms for SRH information and services in rural settings in LMICs.

## Methods

This systematic review is based on the Preferred Reporting Items for Systematic Reviews and Meta-Analyses (PRISMA) guidelines [26]. The protocol was registered in the PROSPERO database on August 21, 2020 (# CRD42020199221).

### Data sources and search strategy

A systematic search of five online databases: Medline, Embase, CINAHL, PsycInfo and Scopus for peer-reviewed articles that reported on mHealth intervention studies with young men and women aged between 10–24 years ('adolescents' aged 10–19 years and 'youth' aged 15–24 years) [27]. These two overlapping age groups were combined to describe 'young people' [28] from rural settings in LMICs. LMICs within this study were determined according to World Bank classifications ([29]. Studies were limited to those published in English from 2000–2023 given the global inception of evidence on mobile telemedicine [30] and increased interest in their use in LMICs over this period. In addition to these sources, reference lists of identified articles and key references of relevant systematic reviews were searched. Searches were also

**Table 1. Search terms and keywords.**

| Terms | |
|---|---|
| **Population of interest** | men, women, male boys, young men, young males, adolescent young men, adolescent boys, men and women, male boys and female girls, young men and women, young males and females, adolescent young men and women, adolescent boys and girls, aged 10–24 years |
| **Intervention (mHealth)** | Mobile health, mHealth, mhealth, m-health, ehealth, e-health, telemedicine, multimedia, cell phone, mobile phone, smart phone, social media |
| **mHealth platforms** | short message services, SMS, text message, interactive voice response (IVR), voice calls, voice services, instant messaging, voice messaging, phone calls. |
| **Intervention platform services** | reproductive health, sexual health, contraception, family planning methods, family planning services, family planning information, sexually transmitted infections, and HIV |
| **Geographical setting** | developing countries, developing economy, less developed countries, underdeveloped countries, LMICs, low-income economy, low-income countries |

carried out in Google and Google Scholar databases. For database searches, medical subject headings (MeSH) were used to identify relevant articles. Key search terms and keywords that were used are shown in Table 1.

The first author (ASL) developed the search strategy which was reviewed by MLH and DL. The search strategy was refined in consultation with the College of Health, Medicine and Well-being's librarian (see Table 1). The Medline search strategy used (see Table 2). Search strategies tailored to the other databases (S1 Table).

## Inclusion and exclusion criteria

We included primary studies that used qualitative methods and reported on the views and experiences of young rural female and male's use of mobile phone platforms for accessing SRH information and services in LMICs. For the purposes of this review, 'young' people are defined as (adolescents and youth populations from 10 to 24 years). Studies in which the ages of the participants were outside this range but could be segregated were included. Conference presentations, student theses, editorials, review articles, letters to the editor, commentaries, symposium proceedings and studies where information on young people could not be disaggregated were excluded.

## Data extraction

Database searches were completed by ASL. The literature results were first downloaded into Endnote X9 software (Thomson Reuters, Scientific Inc., New York City, New York, USA) and

**Table 2. Medline search strategy.**

young* OR youth* OR adolescent* OR young people* OR youth population* OR young women* OR young girl* OR young boy* OR young men* OR young women* OR emerging adult* OR teen* OR adolescent girl* OR adolescent boy* OR adolescen* OR reproductive health* OR sexual health* OR HIV* OR contraception* OR contraceptives* OR modern contraception* OR contracept services* OR contraception counselling* OR family planning* OR family planning services* OR family planning information* OR family education* OR family planning counselling
AND
Healthcare providers* OR Health professionals* OR frontline worker* OR lay worker* OR health provider* OR health counsellor* OR health educator* OR mobile phone health* OR mobile health* OR mHealth* or short message services* OR SMS* OR text message* OR interactive voice response* OR voice calls* OR phone calls*OR voice reminders* OR interactive voice reminders* OR interactive voice calls* OR electronic health* OR phone calls
AND
low-income countries* or low- and-middle-income nation* or low to middle income countries* or middle-income countr* OR developing countries* OR low resource countries* OR Africa* OR sub-Saharan Africa (limit to (english language and full text and humans and yr = "2000–2023").

**Table 3. Definitions of GRADE-CERQual methodological components.**

| Component | Definition |
|---|---|
| Methodological limitations of included studies [32] | The extent to which there are problems in the design or conduct of the primary studies that contributed evidence to a review finding. |
| Coherence of the review finding [33] | An assessment of how clear and cogent the fit is between the data from the primary studies and a review finding that synthesis those data. By cogent in this definition, we mean well supported or compelling. |
| Adequacy of data contributing to review finding [34] | An overall determination of the degree of richness and quantity of data supporting a review finding. |
| Relevance of the included studies to the review question [35] | The extent to which the body of evidence from the primary studies supporting a review finding is applicable to the context (i.e perspective or population, the phenomenon of interest, setting) specified in the review question. |

later imported into the Covidence online platform for analysis. Duplicates were removed and screening was then independently conducted by two reviewers (ASL and NK) based on title and abstract according to the inclusion criteria. Full text articles were retrieved and then two reviewers (ASL and NK) independently assessed the relevancy of the papers. Screened articles were read in full. Discrepancies were resolved by discussion.

## Assessment of methodological quality of studies using GRADE-CERQual approach

The methodological quality of the included studies were appraised using the Grading of Recommendations, Assessment, Development and Evaluation (GRADE) and Confidence in Evidence from Reviews of Qualitative research (CERQual) approach [31]. The methodological quality of included studies were appraised independently by two reviewers (ASL and NK) against four key components: methodological limitations [32], coherence of the review finding [33], data adequacy, [34] and relevance [35] detailed in Table 3 (see also Assessment of Quality in S2 Table). The overall CERQual explanation of quality assessment findings is also detailed in (S2 Table).

## Assessment of studies under CERQual components

All the studies were assessed under each CERQual component in S3 Table.

## Determining the level of confidence in the study finding

We further assessed the level of rating of confidence or certainty in the study findings using the GRADE-CERQual approach in Table 4.

**Table 4. Grading and rating of confidence in the individual review findings.**

| | |
|---|---|
| High confidence | It is highly likely that the review finding is a reasonable representation of the phenomenon of interest. |
| Moderate confidence | It is likely that the review finding is a reasonable representation of the phenomenon of interest |
| Low confidence | It is possible that the review finding is a reasonable representation of the phenomenon of interest |
| Very low confidence | It is not clear whether the review finding is a reasonable representation of the phenomenon of interest |

## Results

### Summary

A total of 3,121 articles were identified through the database search. Among these, 434 duplicates were removed. A total of 2,712 studies were assessed based on titles and abstracts. Of these, 2,672 studies were excluded for not meeting inclusion criteria. Forty full-text articles were assessed for eligibility. Of these, 14 articles were excluded. Twenty-six articles met the inclusion criteria for this review (Fig 1).

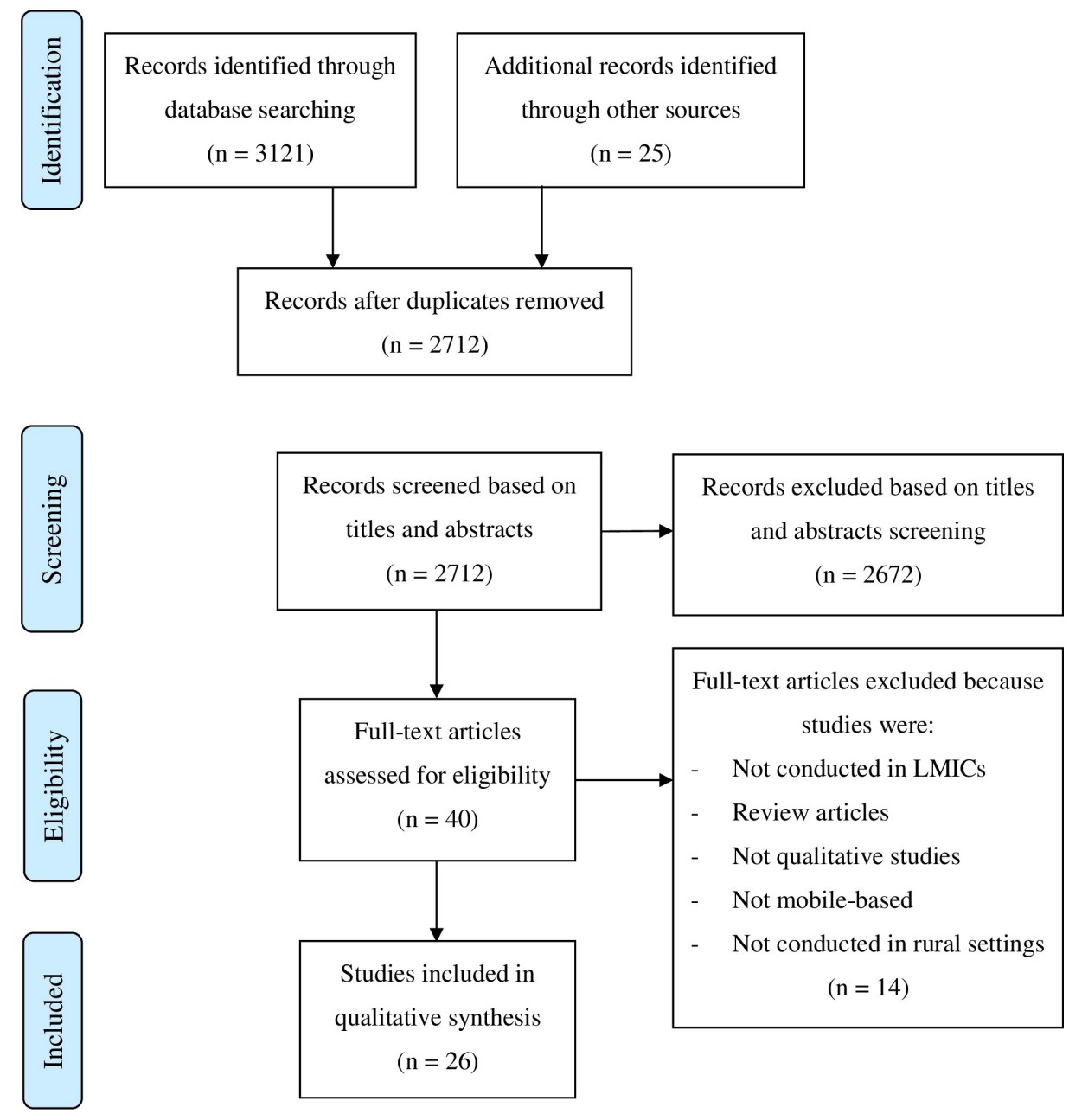

**Fig 1. PRISMA flow chart showing study selection procedure.**

## Characteristics of studies

All the included studies were conducted in rural areas in LMICs and focused on the use of mobile phone platforms for providing access to SRH information and services to young people, including contraceptive advice and information, family planning and HIV prevention education. The studies were conducted in sixteen different countries, with the majority in Kenya (8 studies) and South Africa (3 studies). Two studies were conducted in Nigeria, with an additional four in Uganda (2 studies) and Tanzania (2 studies). The remaining nine studies were conducted in Burkina-Faso, Bangladesh, Malawi (Rwanda, Malawi, and India), Cambodia, Nepal (Tajikistan, Bolivia, and Palestine), Ethiopia, and Peru, respectively. In terms of the study settings, most studies recruited youth participants from communities (17 studies). Others used health care facilities (2), health care facilities and communities (2), communities and schools (3) and schools alone (2). Regarding the methods used for data collection, most involved focus groups (13 studies) and in-depth interviews (9), with four using both (see Table 5).

The 26 studies included in this synthesis used a range of mobile phone platforms for providing access to SRH information and services among young people in rural settings in LMICs. Most of the studies used text-messaging platforms (11) with a combination of voice messaging, (2) interactive voice response, (2) phone calls, Facebook, WhatsApp, and Twitter as noted in Table 5. The majority of studies included information about the barriers to mHealth SRH service access, these have also been included in summary form in Table 5.

## Confidence in the review findings

Confidence in the review findings across studies with explanation of the assessment are detailed in the CERQual evidence profile in Table 6. The study found high confidence in the findings for the text-messaging platforms [15,16,36–39,41,43–53,56,57,59]. However, the confidence in the findings for social media platforms including WhatsApp [15,16,40] Facebook [15,16,40,49] and Twitter [15,49] was low. Generally, the study findings for confidence in the challenges and barriers for platform services was high.

## Preference and acceptability for using mHealth SRH services

In this review, the studies used seven different mobile phone platforms to deliver a range of SRH information and services on contraception, family planning, prevention of STIs and HIV education across rural areas. Participants' acceptance and preference levels for using mobile platform SRH intervention services were varied. Participants reported preferences for SRH text-messaging [15,16,36–39,41,43–53,56,57,59] services. Some of the reasons were associated with their ability to receive text messages even during poor or weak network connectivity conditions, the opportunity to store text messages for future reference and share the messages with friends and peers. Participants also reported a preference for using voice messaging services [42,43,47,52,55]. Some of the reasons were due to their ability to be able to communicate with HCPs without writing, as well as the ability to store voice messages and re-play and listen to them when they needed. Recipients also reported that their ability to interact with HCPs or counsellors to gain quick responses to their SRH issues that were individually tailored to their needs also saw them favour voice messaging types of services over other methods.

The findings indicated that young people preference for using phone call platform to access SRH services from HCPs [39,49,59] was related to their ability to use low-cost basic or non-Android mobile phones as compared to the requirement of expensive Android phone for text messages services which requires the user have literacy or technical skills to be able to compose and read messages.

**Table 5. Summary of studies included in the systematic review.**

| Author(s) | Country & setting | Target population | Data collection method | mHealth delivery platform and services | Key findings on young people views and experiences on barriers for accessing SRH health services |
|---|---|---|---|---|---|
| Adeagbo et al. 2019 [36] | South Africa Community (Rural) | Young females & males aged 18–34 years | Focus groups & In-depth interviews | Text messaging; HIV prevention education | • Limited technological literacy<br>• Cost of data and mobile credit<br>• Cost of phones |
| Akinfaderin-Agarau et al. 2012 [37] | Nigeria Community/ School (Rural/ Urban/ Semi-Urban) | Adolescent girls & young women aged 12–30 years | Focus groups | Text messaging/ Interactive voice response; SRH information. | • Language literacy barrier.<br>• Mobile phone cost and credit<br>• Poor network<br>• Limited power supply<br>• Influences from comm<br>• unity members. Misconceptions and myths on receiving calls or messages from unknown people resulting in death or disappearance<br>• Restriction of use of mobile phone by partners and parents |
| Ampt 2020 [38] | Kenya Community (Rural) | Young women aged 16 years or older. | In-depth interviews | Text messaging; Contraception/ HIV education | • Poor network and stability<br>• Inconvenience for receiving health messages at odd time |
| Blanc et al. 2016 [39] | Nigeria Community/ Schools (Rural/Urban) | Young male & female youth aged 10–24 years | In-depth analysis | Text messaging/ Phone call; SRH information | • Network connectivity<br>• Low literacy in writing and reading text messages due to low education |
| Deliver 2020 [40] | India, Malawi & Rwanda Community (Urban/Semi-Urban/ Rural) | Adolescent girls & young women aged 15–24 years | In-depth interviews | WhatsApp/ Facebook; SRH information | • Influences from communi<br>• ty membersLow technological knowledge<br>• Cost of data and airtime<br>• Language literacy barrier<br>• Cost for smart phones to be able to use services<br>• Lack of expert moderation of Facebook services |
| Dev et al. 2019 [41] | Kenya Health facility (Rural/ Urban) | Young women 14–24 years | In-depth interviews | Text messaging; Contraception/ Family planning | • Low or lack of technical skills in using text messaging platform services<br>• Low technological literacy<br>• Cost of mobile phones |
| Duclos et al. 2017 [42] | Burkina-Faso Community (Rural) | Women | Focus group discussions | Voice messaging; SRH information | • Low technological knowledge on using mobile phones<br>• Restriction of phone use by community members for SRH services<br>• Not able to afford mobile credit & data<br>• Limited mobile network connectivity & stability<br>• Limited electricity supply & stability |
| Eckersberger et al. 2017 [43] | Bangladesh Facility (Urban/Peri-Urban) | Women aged 18 & above | In-depth interviews | Text messaging /Interactive Voice response; Family planning services | • Low technological skills and knowledge<br>• Low formal education/literacy |
| Evelia et al. 2015 [15] | Kenya Community (Rural/ Urban) | Young women & youth aged 10–24 years | Focus group discussions | Text messaging/ Facebook/ WhatsApp; SRH information | • Limited network connectivity and stability<br>• Limited power connectivity and stability associated with cost for charging phones<br>• Not owning mobile phones due to cost<br>• Participants faced restriction of use of mobile phones by community members such as parents, partners, and school authorities for SRH services<br>• WhatsApp, Facebook, and Twitter platforms services were not commonly used by young people due to lack of technical knowledge<br>• Participants think that Facebook platform services lacked expert moderation<br>• Cost of using Facebook, Twitter, and WhatsApp platform services due to the requirement for smart phones<br>• Not having technical skills or knowledge for tweeting<br>• Lack of expert moderation of Facebook platforms services |

*(Continued)*

**Table 5.** (Continued)

| Author(s) | Country & setting | Target population | Data collection method | mHealth delivery platform and services | Key findings on young people views and experiences on barriers for accessing SRH health services |
|---|---|---|---|---|---|
| FHI 360 et al. 2013 [44] | Kenya & Tanzania Community (Rural/Urban) | Young females & males aged 29 years or younger | In-depth telephone interviews | Text messaging; Contraception information | • Cost of mobile phone services |
| Guerrero et al. 2020 [45] | Peru Community (Peri-Urban/Urban) | Adolescents, youth females & males aged 13–24 years | Focus groups | Text messaging; SRH information | • Language literacy barrier<br>• Platform not been gender-based for males and females separately |
| Jamison et al. 2013 [46] | Uganda Community (Rural) | Young male & female aged 18–35 years | Focus groups & in-depth interviews | Text-messaging; Family planning/ HIV prevention education | • Using platform information encourages promiscuity |
| Laidlaw et al. 2017 [47] | Malawi Community/School (Rural) | Male & female youth aged 15–24 years | Focus group discussions | Text messaging/ Interactive Voice Response (IVR); Contraception/ Family planning/ HIV education | • Reliance on borrowed or shared phones<br>• Cost of mobile phones |
| L'engle et al, 2013 [48] | Tanzania Health facility/ Community (Rural/Urban) | Young female & male aged 19–29 years | In-depth interviews | Interactive voice response/ Text messaging; Family planning | • mHealth platforms services devoid of face-to-face interactions but important<br>• Lack of privacy and anonymity |
| McCharty et al. 2018 [16] | Tajikistan, Bolivia & Palestine Community (Rural/Urban) | Young females & males aged 15–30 years | Focus groups | Facebook/ WhatsApp/ Text messaging; Contraception services | • Cost for smart phones<br>• Cost of data and airtime<br>• Limited network connectivity<br>• Limited electricity connectivity |
| Memiah et al. 2014 [49] | Kenya Community (Rural/Urban) | Youth, boys & girls aged 14–29 years | Focus groups | Text messaging/ Phone call/ Facebook/ Twitter; HIV & STIs prevention education | • Cost of mobile/smart phones<br>• Cost of mobile data and credit |
| Merrill et al. 2018 [50] | South Africa Primary Schools (Rural/Urban) | Adolescent girls aged 11–16 years | Focus groups | Text messaging; HIV & STIs education | • Cost of mobile phones<br>• Cost of mobile airtime or credit |
| Nigatu 2017 [51] | Ethiopia Schools (Rural/Urban) | Young women & men Aged 15–24 years | Focus groups & In-depth interviews | Text messaging; HIV & STIs education | • Poor mobile network connectivity and stability |
| Ong et al. 2020 [52] | Cambodia Community (Urban/Rural) | Young females aged 15–24 years | Focus groups | Text messaging/ Voice messaging; HIV/STIs prevention education | • Limited network coverage and reliability<br>• Limited electricity coverage and stability |
| Parajuli & Doneys 2017 [53] | Nepal Community (Rural) | Women & girls aged 16–36 years | Focus groups & in-depth interviews | Text messaging; SRH information | • Not owning mobile phone due to cost<br>• Inability to read and write text messages due to low literacy |
| Sabben et al. 2019 [54] | Kenya Community (Rural/Urban) | Young males & females aged 11 to 14 years | Focus groups | Interactive voice response; HIV prevention education | • Not owning phones<br>• Cost of mobile phones |
| Smith et al. 2017 [55] | Cambodia Community/Health facility (Rural/Urban) | Women aged 22 years & above | In-depth interviews | Voice messaging; Family planning | • Intrusiveness of voice messages<br>• Low technological knowledge |
| Vahdat et al. 2013 [56] | Kenya Community (Rural/Urban) | Males & females aged 19–29 years | In-depth interviews | Short message service (SMS)/ Text-messaging; Contraception information | • Cost of mobile credit<br>• Poor network<br>• Language literacy |
| Visser et al. 2020 [57] | South Africa Community (Urban/Rural) | Young people (12–24 years | Focus groups | Interactive voice response/ Text messaging; HIV prevention education | • Cost of mobile airtime or data<br>• Limited network access and stability |

(*Continued*)

**Table 5.** (Continued)

| Author(s) | Country & setting | Target population | Data collection method | mHealth delivery platform and services | Key findings on young people views and experiences on barriers for accessing SRH health services |
|---|---|---|---|---|---|
| Winskell et al. 2018 [58] | Kenya Community (Peri-Urban) | Young females & males aged 11–14 years | Focus groups | Interactive platform; Contraception services | • Lack of technological knowledge <br> • Lack of technological literacy skills |
| Ybarra et al. 2020 [59] | Uganda Community (Rural/Urban) | Youth (women & men) aged 18- to 22 years | Focus groups | Phone call/ Text messaging; HIV prevention education | • Cost for phone credit or airtime <br> • Women faced pressure from partners to share text messages |

**Note:** The findings of this study relate to young people aged between 10–24 years were extracted from the articles. The findings of this study relate to young people aged between 10–24 years were extracted from the articles.

### Barriers to using mHealth platforms SRH services

Barriers to using mobile platform services in rural settings centred on cost for mobile phones and their maintenance in terms of mobile credit or airtime and data, technological barriers related to lack of technological knowledge or skills needed to use platform services [15,36,37,39–43,45,53,55,56,60], limited infrastructure in terms of electricity and network connectivity and stability, and influence of social and cultural norms, influences from community members and beliefs and perceptions [15,37,40,42,46,59], and the restriction of mobile phone use especially in instances where young people require parental consent to be able to use mobile phones for SRH services and pressure from partners of young women to share text messages or phone conversations with them imposed constraints and compromised privacy and confidentiality [15,37,40,42,47,59] as indicated in Table 2.

### Discussion

This review qualitatively explored the experiences and perceptions of young peoples' use of mobile phone platforms for SRH information and services in rural settings in LMICs using the GRADE-CERQual approach. Evidence has shown that mHealth technology platforms have the potential to address numerous SRH information and services access barriers for young people in rural contexts in LMICs [21,23,25,61–63]. The study findings reported preferences and barriers for use of mHealth-based platforms for SRH information and services among rural young people across rural LMICs.

Generally, cost in terms of mobile phone ownership and credit or data for their maintenance was reported as major barrier which influenced use of platforms SRH services among young people [15,16,36,37,40–42,44,47,49,50,53,54,56,57,59]. The cost factor led to the reliance on borrowed or shared phones by people who could not afford to be able to access SRH services. The use of borrowed and shared phones resulting in privacy and confidentiality challenges was also reported in this study [15,37,40,42,47,59] with high confidence in the study findings noted. Mobile phone sharing is a common phenomenon across rural settings in LMICs among poor young people and creates potential source of privacy and confidentiality breaches [64–67]. This is a key challenge when discussing SRH information and services especially when concerning sensitive issues [37,68]. As such, young people are reluctant to use borrowed or shared phone for sensitive SRH issues [37]. Ownership of a mobile phone guarantees independence and freedom for use for SRH information and services [37]. However, as rural young people are not economically independent, they may not be able to acquire and maintain

**Table 6. CERQual evidence profile.**

| Summary of review findings | Studies contributing to review findings | CERQual assessment of confidence in the evidence | Explanation of CERQual assessment |
|---|---|---|---|
| **General experiences and perceptions for SRH mHealth platforms SRH information and services** Participants had a wide range of experiences and views regarding interest, acceptability, and preferences for receiving mobile phone based SRH information and services. These were centred on comfort, convenience, privacy, confidentiality, anonymity, non-judgemental, ability for interaction, affordability and easier for accessing SRH messages compared to face-to-face services where communities are far away from health facilities reported across rural settings in LMICs. | [15,16,36–59] | High confidence | It is highly likely that the review finding is a reasonable representation of participants experiences for using mobile platforms SRH information and services. For the 26 studies, 3 had unclear ethical procedures, 3 had unclear reflexivity, 1 had unclear data analysis procedure and 3 with minor data issues. |
| **Experiences and perceptions for SRH mHealth platform SRH services** | | | |
| **Text messaging** Participants reported their preference for text messaging services due to the ability to store text messages for reference and share with friends and peers. Some participants also indicated being able to receive text messages under poor or weak network connectivity conditions as positive experiences for using the service. | [15,16,36–39,41,43–53,56,57,59] | High confidence | It is highly likely that the finding is a reasonable representation of participants experiences for text messaging. For the 21 studies, 2 had unclear ethical procedures, 1 an unclear data analysis procedure and 3 with minor data issues. |
| **Voice messaging** Participant's ability to store voice messages and re-play and listen at their own convenience was expressed by some participants. Some participants with low-literacy or no formal education expressed their preference for the service as it did not require writing and reading of text messages. | [42,43,47,52,55] | High confidence | It is highly likely that the finding is a reasonable representation of participants experiences for voice messaging. All the 5 studies had no or very minor concerns. |
| **Interactive voice response** Most participants expressed positive views and preferences for interactive voice response due to the ability to interact or engage with health care providers or counsellors to discuss issues that were tailored to their needs. Young people with low education also indicated the ease of using the platform. The requirement for a basic mobile phone to use the services was reported as very advantageous to young poor rural people. | [37,43,48,50,55–57] | High confidence | It is highly likely that the finding is a reasonable representation of participants experiences for interactive voice response platforms. For all the 7 studies, one had minor data issues. |
| **Mobile phone call** Participants reported positive experiences due to the ability to interact with health providers and to ask personal questions and get an instant response or feedback tailored to their needs. Low-literate young people liked the services for not requiring composing or reading text messages and not requiring smart or android phones to use the SRH intervention services. | [39,49,59] | Moderate confidence | It is possible that the finding is a reasonable representation of participants experiences for mobile phone call platform intervention SRH information and services. For all the 3 studies, ethical procedures was not clearly stated in one study. |
| **WhatsApp** Participants reported positive experiences in terms of ability to interact with providers and friends and posting or sharing of questions on their health issues on the platform for answers or responses. However, some expressed the services not been affordable due to the need for an android or smart phone to be able to use services as a challenge. | [15,16,40] | Low confidence | It is possible that the finding is a reasonable representation of participants experiences for WhatsApp platform SRH information and services due to limited data. For all the 3 studies, ethical procedures in one of the studies were not clearly stated. |

(*Continued*)

**Table 6.** (Continued)

| Summary of review findings | Studies contributing to review findings | CERQual assessment of confidence in the evidence | Explanation of CERQual assessment |
|---|---|---|---|
| **Facebook**<br>Participants who used the services, a preference was related to the ability to interact with friends and peers and posting questions to receive answers or responses. Some participants reported that the requirement for an android and smart phone to be able to use the services was a barrier. Some had the perception that the services lacked expert moderation of the platform and information posted may not be authentic. | [15,16,40,49] | Very low confidence | It is not clear whether the finding is a reasonable representation for Facebook platform SRH services due to thin data. However, for the 4 studies, the ethics procedures in one were unclear. |
| **Twitter**<br>Participants who used twitter platform services reported not being conversant with the platform services They also indicated that it familiar among the youth in their settings. Participants said they encountered some technical difficulties in manipulating the twitter handle. The need for an android or smart phone to be able to use services was also cited as a challenge. | [15,49] | Very low confidence | It is not clear whether the finding is a reasonable representation of participants experiences for Twitter platform SRH services due to thin data. For the 2 studies, one had unclear ethics procedures. |
| **General barriers for use of mobile platforms SRH information and services by participants** | | | |
| **Cost**<br>Participants cited cost for mobile phones and cost for mobile credit or airtime and data as a barrier to use of platforms services. In some instances, some had to depend on shared or borrowed phones to be able to access platforms services. Cost barrier was widely reported across several study settings. | [15,16,36,37,40–42,44,47,49,50,53,54,56,57,59] | High confidence | It is highly likely that the finding is a reasonable representation of participants experiences of barriers in using SRH mobile phone platforms services. For all the 16 studies, one study had unclear ethical procedures, two with thin data and one with unclear reflexivity issues. |
| **Technological skills and language literacy**<br>Low technological knowledge, skills and language literacy were cited as barriers influencing fully use mobile phone platforms SRH services in across several rural settings | [15,36,37,39–43,45,53,55,56,60] | High confidence | It is highly likely that the finding is a reasonable representation of participants experiences of barriers in using SRH mobile phone platforms services. For all the 13 studies, two of the studies had unclear ethics procedures and unclear reflexivity. |
| **Electricity connectivity and stability**<br>Infrastructural barriers related to limited electricity connectivity and stability was cited a barrier affecting use of SRH mobile phone platforms services across several rural settings. This barrier was commonly reported across several rural settings. | [15,16,37,42,52] | High confidence | It is highly likely that the finding is a reasonable representation of participants experiences for mobile phone platforms SRH information and services. For all the 5 studies, one had unclear ethics procedures. |
| **Network connectivity and stability**<br>Infrastructural barriers related to limited and weak network connectivity and stability was cited a barrier affecting use of SRH mobile phone platforms services across rural settings. This barrier was commonly reported across several rural settings. | [15,16,37–39,42,51,52,56,57] | High confidence | It is highly likely that the finding is a reasonable representation of participants experiences for SRH mHealth platforms SRH information and services. For all the 10 studies, 3 had unclear ethics procedure issues, one unclear reflexivity, and 2 with thin data. |
| **Privacy and Confidentiality**<br>Community members such as parents, partners/husbands and school authorities played a major role in influencing young people use of mobile phones to access SRH information and services in rural settings. The restriction of mobile phone use especially in instances where young people require parental consent to be able to use mobile phones for SRH services and pressure from partners for young women to share text messages or phone conversations with them imposed constraints and compromising privacy and confidentiality. | [15,37,40,42,47,59] | High confidence | It is highly likely that the finding is a reasonable representation of participants experiences for SRH mHealth platforms SRH information and services. For all the 6 studies, 1 had unclear ethics procedure issues and 1 minor data issue, |

*(Continued)*

**Table 6.** (Continued)

| Summary of review findings | Studies contributing to review findings | CERQual assessment of confidence in the evidence | Explanation of CERQual assessment |
|---|---|---|---|
| **Social norms and beliefs**<br>Young people faced social and cultural norms barriers because of influences from community members in the rural communities. In these rural settings socio-cultural norms and religious beliefs are ingrained SRH services use. Community members such parents and partners referred to young women using mHealth platforms for SRH services as been promiscuous. Some young people had the belief that receiving phone calls from unknown mobile phone contact numbers or persons could lead to death or the disappearance of the recipient. These restrictive beliefs and myths were reported across Tajikistan, Bolivia, Palestine, and Nigeria. | [15,37,40,42,46,59] | Moderate confidence | It is highly likely that the finding is a reasonable representation of participants experiences for use of mobile phone platforms intervention SRH information and services. For all the 6 studies, 2 studies had issues with one was unclear reflexivity and the other on unclear ethical procedures. |

mobile phones on their own for SRH information and services without financial interventions. Introduction of financial interventions is needed to make mobile platforms SRH services affordable for poor rural young people [68]. Introducing financial subsidies for the cost of mobile phones and airtime for maintenance through flexible terms of payment could help overcome the cost barrier and sustain use of services among young people in rural settings in LMICs [69].

The review findings reported limited and poor network [15,16,37–39,42,51,52,56,57] and electricity, connectivity, and coverage [15,16,37,42,52] as barriers to use of mobile platforms SRH information and services among young people across the study settings. This is consistent with other studies which have identified limited coverage and poor network and power connectivity as a challenge in rural LMICs to the access of SRH information and services with young people struggling to identify spots in the communities with network connectivity and stability to be able to use mobile phones [68,70–73]. In addition, charging phones often requires travel to communities with power and sometimes paying a fee to charge mobile phones batteries. Given that the use of mHealth applications in rural settings requires a reliable, effective, and sustained network and electricity connectivity [22,74], exploring alternative power sources such as solar panels and power banks have been recommended for rural poor-resourced settings in LMICs with limited technological infrastructure [22,69].

High confidence was found in studies reporting on limited technological skills and language literacy [15,36,37,39–43,45,53,55,56,60] as barriers to mHealth for SRH particularly in terms of limited knowledge regarding the use of these platforms. Despite the perceived benefits and opportunities of mobile phone technology for SRH information services among rural populations (especially young people in rural communities in LMICs), these services are less utilisable by low literate and illiterate young people in rural contexts that would otherwise have benefited most from mHealth technologies SRH initiatives [75].

Moderate confidence was shown in the findings that identified the restriction of young people's mobile phone use for SRH services by community members such as partners and parents [15,37,40,42,46,59] as a barrier to use. This finding is consistent with other studies in rural LMICs where young women reported that their sexual partners associated phone use for SRH information with infidelity [66,67,70]. Despite the untapped potential of mobile phones as tools for improving access to SRH information and services among young people in rural LMICs, the use of mobile phones frequently raise a topic of conflict between sexual partners

[70] and could influence the decision of young people to use innovative mHealth technology for SRH information and services. Other perceived factors which influenced young people's phone use for SRH services emanated from social and beliefs norms and myths related to phone calls received from unknown people that could result in death or disappearance of the recipient. This was most often reported across rural settings in Tajikistan, Bolivia, Palestine, and Nigeria. Other studies have also linked social beliefs and myths for receiving calls from anonymous people resulting in death caused by "evil spirits" [66,67]. The cultural and religious context plays an essential role, as community members such as parents may hold reservations about sexual communication among rural young population [12–14]. Socio-cultural norms might influence communication and restrict community members interaction and communication with young peoples on their decision-making regarding SRH services uptake [12–14]. These socio-cultural norms and beliefs need to be addressed to avoid misinforming young people for using the mobile phone technology for SRH information and services in rural context of LMICs. Public education is critical to educate rural populations about mHealth SRH initiatives to address knowledge gaps in relation to myths and perceptions that affect use of mobile phone platforms for SRH information and services among young people in rural contexts in LMICs [76, 77].

The study also reported positive findings regarding the acceptance and preferences of mHealth platforms SRH information and services in relation to youth-friendly platforms services [23,78,79] associated with comfort, convenience, privacy and confidentiality, compared to in-person services [15,16,36–59]. The ability to engage young people with a wide range of SRH information and services across wide geographical rural areas were reported as some of the benefits of mHealth platforms for SRH services [15,16,36–59]. Several studies have come out with similar findings among young people in rural contexts in LMICs [6,80–83]. With regards to platforms such as text messaging, voice messaging, integrated voice response (IVR) and phone calls, some of the positive findings in terms of their preference for SRH information and services was due to their familiarity and the ability to use low-cost non-Android phones for the services, compared to WhatsApp, Facebook, and Twitter platforms. The preference for text-messaging services was associated with the ability to receive and read text messages under poor or weak network conditions and the opportunity to store messages for reference and share with friends and peers [15,16,36–39,41,43–53,56,57,59]. These findings are also confirmed by studies in rural settings in LMICs among young people [6,45,68,84–86]. Text-messaging platforms continue to be the most frequently used and preferred mobile phone communication platform among populations in rural settings in LMICs [37]. However, rural young peoples' preference for phone call [39,49,59], voice messaging [42,43,47,52,55] and IVR [37,43,48,50,55–57] platforms SRH services was due to not requiring literacy skills to be able compose or read messages [42,43,47,52,55] as compared to text messaging services. Also, young people's ability to interact or engage with HCPs or counsellors to discuss their SRH issues and receiving prompt response tailored to their needs were also reported as some the reasons for their preference [37,43,48,50,55–57].

While basic or simple low cost mobile phone ownership levels are common among rural young people in LMICs and could be used for mHealth SRH initiatives, the same is not true for smartphones [87,88]. While mobile phone-based platforms are becoming increasingly popular among young people for SRH services in rural contexts in LMICs, platforms such as WhatsApp, Facebook and Twitter are less used and preferred by young people [16,40,49]. Few young people had used these platforms due to the requirement for smart phones and their associated high cost and lack of technological skills [21,89,90]. Some participants also reported that the use of WhatsApp, Facebook, and Twitter platforms SRH services were not familiar among them in their communities. Evidence has shown that there is a dearth of qualitative

evidence on the effect of innovative social media platforms such WhatsApp, Facebook, and Twitter for SRH information and services among young people in rural LMICs [16,40,49]. This warrants further research on these social media platforms to help maximise their impact for use for SRH information and services among young people in rural LMICs settings.

We acknowledge that this review has some limitations. Firstly, we restricted our search to those studies published in the English language between 2000–2023. This could have limited the numbers of articles restricted from the search. We could have missed relevant articles outside of our study period and languages other than English. Also, for not including genders, such as transgender and non-binary young people, could be a limitation. Despite the limitations, this study was conducted using a comprehensive search strategy, a robust quality appraisal process and rigorous systematic review methodology [32–35].

## Conclusion

The findings of this synthesis have demonstrated the perceived potential of mHealth platforms for improving access to conventional SRH information and services among young people in rural settings in LMICs. While mHealth platforms have the potential to address access limitations to SRH information and services among rural young people, the barriers need to be addressed. These findings provide relevant recommendations for policy makers and implementers of SRH programs to improve services for integration and scale-up in the rural health system contexts in LMICs.

## Supporting information

**S1 Table. Search strategies tailored to Cochrane Library, Scopus, CINAHL and PsychoINFO.**
(DOCX)

**S2 Table. Overall summary of assessment of quality and CERQual explanation of findings.**
(DOCX)

**S3 Table. Assessment of studies under each CERQual component.**
(DOCX)

## Acknowledgments

We would like to acknowledge Jessica Birchall, senior librarian at the University of Newcastle for her guidance in developing the search terms for this review.

## Author Contributions

**Conceptualization:** Alexander S. Laar, Melissa L. Harris, Deborah Loxton.

**Formal analysis:** Alexander S. Laar, Md N. Khan.

**Methodology:** Alexander S. Laar, Md N. Khan, Deborah Loxton.

**Project administration:** Deborah Loxton.

**Supervision:** Melissa L. Harris, Deborah Loxton.

**Writing – original draft:** Alexander S. Laar.

**Writing – review & editing:** Alexander S. Laar, Melissa L. Harris, Md N. Khan, Deborah Loxton.

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
