## [Decision Letter · Decision Letter 0]

27 Sep 2023

PDIG-D-23-00332

Views and experiences of young people on using mHealth platforms for sexual and reproductive health services in rural low- and middle-income countries: a qualitative systematic review

PLOS Digital Health

Dear Dr. Laar,

Thank you for submitting your manuscript to PLOS Digital Health. After careful consideration, we feel that it has merit but does not fully meet PLOS Digital Health's publication criteria as it currently stands. Therefore, we invite you to submit a revised version of the manuscript that addresses the points raised during the review process.

Please submit your revised manuscript within 60 days Nov 26 2023 11:59PM. If you will need more time than this to complete your revisions, please reply to this message or contact the journal office at digitalhealth@plos.org. Please include the following items when submitting your revised manuscript:

We look forward to receiving your revised manuscript.

Kind regards,

Haleh Ayatollahi

Section Editor

PLOS Digital Health

Journal Requirements:

1. Please provide separate figure files in .tif or .eps format only and remove any figures embedded in your manuscript file. Please also ensure that all files are under our size limit of 10MB.

2. We ask that a manuscript source file is provided at Revision. Please upload your manuscript file as a .doc, .docx, .rtf or .tex.

3. Please ensure that Funding Information and Financial Disclosure Statement are matched.

4. In the Funding Information you indicated that no funding was received. Please revise the Funding Information field to reflect funding received.

5. In the online submission form, you indicated that "Data is available from the corresponding author upon a reasonable request". All PLOS journals now require all data underlying the findings described in their manuscript to be freely available to other researchers, either 1. In a public repository, 2. Within the manuscript itself, or 3. Uploaded as supplementary information.

Additional Editor Comments (if provided):

Reviewers' comments:

Reviewer's Responses to Questions

**Comments to the Author**

1. Does this manuscript meet PLOS Digital Health’s publication criteria? Is the manuscript technically sound, and do the data support the conclusions? The manuscript must describe methodologically and ethically rigorous research with conclusions that are appropriately drawn based on the data presented.

Reviewer #1: Yes

Reviewer #2: Yes

Reviewer #3: Yes

2. Has the statistical analysis been performed appropriately and rigorously?

Reviewer #1: N/A

Reviewer #2: N/A

Reviewer #3: N/A

3. Have the authors made all data underlying the findings in their manuscript fully available (please refer to the Data Availability Statement at the start of the manuscript PDF file)?

Reviewer #1: No

Reviewer #2: Yes

Reviewer #3: Yes

4. Is the manuscript presented in an intelligible fashion and written in standard English?

Reviewer #1: Yes

Reviewer #2: Yes

Reviewer #3: Yes

5. Review Comments to the Author

Reviewer #1: The study is a review that “qualitatively explored the experiences and perceptions of young people’s use of mobile phone platforms for SRH information and services” (lines 13-15). The manuscript is interesting, generally well-written and easy to follow. Alas, I have some concerns with the manuscript in its current state.

In the PROSPERO protocol, the research question is “How effective are mHealth communication platforms/channels for accessing sexual and reproductive health information and services by young people in rural settings in LMICs?” (CRD42020199221) whereas in the manuscript the authors “synthesized the current evidence to provide a comprehensive understanding of the views and experiences of young people in rural areas using mobile platforms for SRH information and services in rural and remote areas in LMICs.” The authors should update the protocol and provide a rationale for the change of focus from effectiveness to experiences in the manuscript. They should also state the current research question more clearly at the end of the introduction.

The PROSPERO protocol lists the following databases for the searches: MEDLINE, Embase, PsycINFO and Scopus. In the manuscript an additional database is listed (CINAHL). The authors should revise the protocol and explain all deviations from the original plan in the manuscript.

Regarding population of interest, the authors only mention boys/men and girls/women (page 9). A rationale for not including more genders, such as transgender and non-binary, should be provided – and discussed as a limitation to the study in the discussion. The logic behind only specifying “Africa” and “sub-Saharan Africa” in the search terms should also be given. Thirdly, the target age range was 10-24 (page 9), but it is clear from Table 5 that the authors have included studies with older participants:

- Vahdat: 19-29 years

- Ducleos: age range?

- Smith: 22 years & above

- Jamison: 18-35

- Parajuli & Doneys: 16-36

- McCharty: 15-30

- Adeagbo: 18-34

- Akinfaderin-Agarau: 12-30

- Ampt: 16 years or older

- FHI 360 USAID (?): 29 years or younger

- Memiah: 14-29 years

- Eckersberger: 18 & above

- L’engle: 19-29 years

I also suggest that the authors sort the included articles in Table 5 in alphabetical order and provide year of publication after respective authors' names. Furthermore, the results column is framed as “Key findings on barriers for young people in accessing SRH health”, which only partly reflects the research question (views and experiences).

Last but not least, the search was restricted to the years 2000-2020. It is of utmost importance that the authors include potential studies published between 2020 up until today, especially considering the rapid development of mHealth interventions, why the authors must update the search.

Reviewer #2: A rigorous, and well written paper was submitted to the journal. The authors abided to the requirements of a qualitative systematic review. I cannot fault them for this paper in terms of method, or writing.

Reviewer #3: The authors have successfully examined several important factors impacting access to SRH services among young individuals in LMICs, including cultural context, financial barriers, and privacy and confidentiality concerns by synthesizing available evidence. If possible, another aspect that might be worth discussing is the role of parental or major financial supporters on young people's access to SRH services, given their potential impact in either facilitating or hindering access.

These individuals often play an important role either by providing the necessary financial resources for obtaining a phone and services or by imposing constraints, such as sharing a phone among family members, which could potentially compromise confidentiality, as pointed out in the review. 

Moreover, the cultural context plays an essential role, as parents may hold reservations about sexual contact among this population due to religious or cultural reasons, potentially leading to concerns that if parents were aware of such activities, they might resort to actions that can influence young people's decision-making regarding SRH services.

Exploring these multifaceted dynamics could provide a more comprehensive understanding of the challenges and opportunities in this context. If data regarding these aspects is lacking or inadequate, acknowledging this limitation in the discussion section would enhance the manuscript.

6. PLOS authors have the option to publish the peer review history of their article (what does this mean?). If published, this will include your full peer review and any attached files.

**Do you want your identity to be public for this peer review?** For information about this choice, including consent withdrawal, please see our Privacy Policy.

Reviewer #1: Yes: Siri Jakobsson Støre

Reviewer #2: No

Reviewer #3: No

---

## [Decision Letter · Decision Letter 1]

8 Jul 2024

PDIG-D-23-00332R1

Views and experiences of young people on using mHealth platforms for sexual and reproductive health services in rural low- and middle-income countries: a qualitative systematic review

PLOS Digital Health

Dear Dr. Laar,

Thank you for submitting your manuscript to PLOS Digital Health. After careful consideration, we feel that it has merit but does not fully meet PLOS Digital Health's publication criteria as it currently stands. Therefore, we invite you to submit a revised version of the manuscript that addresses the points raised during the review process.

Please submit your revised manuscript within 30 days Aug 07 2024 11:59PM. If you will need more time than this to complete your revisions, please reply to this message or contact the journal office at digitalhealth@plos.org. Please include the following items when submitting your revised manuscript:

We look forward to receiving your revised manuscript.

Kind regards,

Haleh Ayatollahi

Section Editor

PLOS Digital Health

Journal Requirements:

2. Please ensure that Funding Information and Financial Disclosure Statement are matched.

3. In the Funding Information you indicated that no funding was received. Please revise the Funding Information field to reflect funding received.

4. We ask that a manuscript source file is provided at Revision. Please upload your manuscript file as a .doc, .docx, .rtf or .tex.

Additional Editor Comments (if provided):

Reviewers' comments:

Reviewer's Responses to Questions

**Comments to the Author**

1. If the authors have adequately addressed your comments raised in a previous round of review and you feel that this manuscript is now acceptable for publication, you may indicate that here to bypass the “Comments to the Author” section, enter your conflict of interest statement in the “Confidential to Editor” section, and submit your "Accept" recommendation.

Reviewer #1: All comments have been addressed

Reviewer #4: All comments have been addressed

Reviewer #5: All comments have been addressed

2. Does this manuscript meet PLOS Digital Health’s publication criteria? Is the manuscript technically sound, and do the data support the conclusions? The manuscript must describe methodologically and ethically rigorous research with conclusions that are appropriately drawn based on the data presented.

Reviewer #1: Yes

Reviewer #4: Yes

Reviewer #5: Yes

3. Has the statistical analysis been performed appropriately and rigorously?

Reviewer #1: N/A

Reviewer #4: N/A

Reviewer #5: N/A

4. Have the authors made all data underlying the findings in their manuscript fully available (please refer to the Data Availability Statement at the start of the manuscript PDF file)?

Reviewer #1: No

Reviewer #4: Yes

Reviewer #5: Yes

5. Is the manuscript presented in an intelligible fashion and written in standard English?

Reviewer #1: Yes

Reviewer #4: Yes

Reviewer #5: Yes

6. Review Comments to the Author

Reviewer #1: The authors have adressed all my previous comments.

Reviewer #4: This paper presents a review of the experiences of young individual living in LMICs in using mobile phone platforms for SRH information and services. The review is thorough, and described well and is relevant to the PLOS Digital Health audience. However, the English in the paper needs to be revised before it can be accepted. Especially in the discussion section I found grammatical mistake in almost every sentence. I started listing errors I found below, but there were too many to list out in a review. Additionally there are sentences that are technically grammatically correct but awkwardly phrased. To be clear, this is a good paper and the English isn't so bad that it's unintelligible, but the frequent grammatical errors and sometimes awkward phrasings make it a less-than-smooth read. I would encourage the authors to have a native English speaker review the paper to help with the clarity or at the very least spend some time themselves going through the paper sentence by sentence to fix the grammar and improve clarity/flow. Although I wasn't able to list out all the issues, I hope that the ones I have called out will help the authors understand the types of issues that need to be addressed systemically.

Methods line 100: The added sentence in green here is completely unnecessary and should be removed. Stating that "Studies were limited to those published in English from 2000-2023 given..." is perfectly sufficient.

Methods line 122: The sentence that starts with "Also, in studies where" should be "Studies in which the ages of the participants were outside this range but could be segregated were included."

Table 5: "et al" should be "et al." That is et al. always ends with a period.

Results line 29: "partners for young women" should be "partners of young women"

Results line 17: This sentence is very awkwardly phrased, and took me several rereads to understand what was being said.

Discussion line 62: missing commas, should be "electricity, connectivity, and coverage"

Discussion line 93: missing comma between Bolivia and Palestine

Discussion line 95: "religion" should be "religious"

Discussion line 116: "non android" should be "non-Android"

Discussion line 126: sentence starting with "Preference for voice messaging..." has several grammatical mistakes. "to use" (line 127) doesn't seem grammatically correct, and "the to the" (line 128) seems like a typo/a word is missing.

Discussion line 138: "Whiles" should be "While"

Discussion line 130: The sentence that begins with "The findings indicated that" seems to be redundant with the entire next paragraph (lines 129-141), and I'm not sure why this redundancy is needed. This sentence is also a bit difficult to parse since there's so much information contained in it, personally I'd recommend getting rid of this sentence since as far as I can tell all its information is contained in the following paragraph. If the authors want to keep the sentence, I'd make it clearer why the redundancy with the next paragraph is needed or rewrite the sentence to reduce redundancy.

Discussion line 143: what does "familiar among them in their settings" mean? This should be rephrased for clarity

Discussion line 153: The green line here is just not grammatically correct. Also, it is a limitation, the sentence shouldn't say it "could be" a limitation.

Discussion line 151: Shouldn't this be 2000-2023?

Discussion line 156: there is no need to mention the strength of the study

Acknowledgments line 182: Shouldn't use "my" in a research paper

Reviewer #5: The manuscript is rigorously written and follows the systematic review guidelines. The methods are sound, and the findings are presented clearly. 

While the introduction does a good job of setting the context, it could benefit from a brief mention of any significant advancements or changes in mHealth interventions post-2020, given the rapid evolution of technology.

 It would be beneficial to elaborate on the role of socio-cultural norms and parental influence, as these are significant barriers. Discussing the potential impact of these factors on the uptake of mHealth services can provide deeper insights.

7. PLOS authors have the option to publish the peer review history of their article (what does this mean?). If published, this will include your full peer review and any attached files.

**Do you want your identity to be public for this peer review?** For information about this choice, including consent withdrawal, please see our Privacy Policy. 

Reviewer #1: Yes: Siri Jakobsson Støre

Reviewer #4: No

Reviewer #5: No

---

## [Decision Letter · Decision Letter 2]

17 Sep 2024

PDIG-D-23-00332R2

Views and experiences of young people on using mHealth platforms for sexual and reproductive health services in rural low- and middle-income countries: a qualitative systematic review

PLOS Digital Health

Dear Dr. Laar,

Thank you for submitting your manuscript to PLOS Digital Health. After careful consideration, we feel that it has merit but does not fully meet PLOS Digital Health's publication criteria as it currently stands. Therefore, we invite you to submit a revised version of the manuscript that addresses the points raised during the review process.

Please submit your revised manuscript within 30 days Oct 17 2024 11:59PM. If you will need more time than this to complete your revisions, please reply to this message or contact the journal office at digitalhealth@plos.org. Please include the following items when submitting your revised manuscript:

We look forward to receiving your revised manuscript.

Kind regards,

Haleh Ayatollahi

Section Editor

PLOS Digital Health

Haleh Ayatollahi

Section Editor

PLOS Digital Health

Journal Requirements:

Additional Editor Comments (if provided):

Reviewers' comments:

Reviewer's Responses to Questions

**Comments to the Author**

1. If the authors have adequately addressed your comments raised in a previous round of review and you feel that this manuscript is now acceptable for publication, you may indicate that here to bypass the “Comments to the Author” section, enter your conflict of interest statement in the “Confidential to Editor” section, and submit your "Accept" recommendation.

Reviewer #4: All comments have been addressed

2. Does this manuscript meet PLOS Digital Health’s publication criteria? Is the manuscript technically sound, and do the data support the conclusions? The manuscript must describe methodologically and ethically rigorous research with conclusions that are appropriately drawn based on the data presented.

Reviewer #4: Yes

3. Has the statistical analysis been performed appropriately and rigorously?

Reviewer #4: N/A

4. Have the authors made all data underlying the findings in their manuscript fully available (please refer to the Data Availability Statement at the start of the manuscript PDF file)?

Reviewer #4: (No Response)

5. Is the manuscript presented in an intelligible fashion and written in standard English?

Reviewer #4: (No Response)

6. Review Comments to the Author

Reviewer #4: While I still think the authors should make another grammar pass before their final submission, this is much better and I'm satisfied with the updates the authors have made. A few things I noticed while reviewing:

1. Discussion Line 63: "The review finding reported..." should be "The review findings reported"

1. Discussion Line 64: "...as a barriers to use of..." should be "...as barriers to the use of..."

2. Discussion Line 113: "The ability to engage young people with wide range..." should needs an "a" before "wide range"

3. Discussion Line 126-127: peoples should be possessive, so this should be "...rural young peoples' preference..."

7. PLOS authors have the option to publish the peer review history of their article (what does this mean?). If published, this will include your full peer review and any attached files.

**Do you want your identity to be public for this peer review?** For information about this choice, including consent withdrawal, please see our Privacy Policy. 

Reviewer #4: No

---

## [Decision Letter · Decision Letter 3]

9 Oct 2024

Views and experiences of young people on using mHealth platforms for sexual and reproductive health services in rural low- and middle-income countries: a qualitative systematic review

PDIG-D-23-00332R3

Dear Mr Laar,

We are pleased to inform you that your manuscript 'Views and experiences of young people on using mHealth platforms for sexual and reproductive health services in rural low- and middle-income countries: a qualitative systematic review' has been provisionally accepted for publication in PLOS Digital Health.

Best regards,

Haleh Ayatollahi

Section Editor

PLOS Digital Health

Reviewer Comments (if any, and for reference):

Reviewer's Responses to Questions

**Comments to the Author**

1. If the authors have adequately addressed your comments raised in a previous round of review and you feel that this manuscript is now acceptable for publication, you may indicate that here to bypass the “Comments to the Author” section, enter your conflict of interest statement in the “Confidential to Editor” section, and submit your "Accept" recommendation.

Reviewer #4: All comments have been addressed

2. Does this manuscript meet PLOS Digital Health’s publication criteria? Is the manuscript technically sound, and do the data support the conclusions? The manuscript must describe methodologically and ethically rigorous research with conclusions that are appropriately drawn based on the data presented.

Reviewer #4: Yes

3. Has the statistical analysis been performed appropriately and rigorously?

Reviewer #4: N/A

4. Have the authors made all data underlying the findings in their manuscript fully available (please refer to the Data Availability Statement at the start of the manuscript PDF file)?

Reviewer #4: Yes

5. Is the manuscript presented in an intelligible fashion and written in standard English?

Reviewer #4: Yes

6. Review Comments to the Author

Reviewer #4: All comments have been addressed.

7. PLOS authors have the option to publish the peer review history of their article (what does this mean?). If published, this will include your full peer review and any attached files.

**Do you want your identity to be public for this peer review?** For information about this choice, including consent withdrawal, please see our Privacy Policy.

Reviewer #4: No
